# Excess mortality by specific causes of deaths in the city of São Paulo, Brazil, during the COVID-19 pandemic

Gisele Aparecida Fernandes[1], Antonio Paulo Nassar Junior[2], Gulnar Azevedo e Silva[3], Diego Feriani[4], Ivan Leonardo Avelino França e Silva[4], Pedro Caruso[2], Maria Paula Curado[1]*

1 Group of Epidemiology and Statistics on Cancer, AC Camargo Cancer Center, São Paulo, SP, Brazil, 2 Intensive Care Unit, AC Camargo Cancer Center, São Paulo, SP, Brazil, 3 Department of Epidemiology, Universidade do Estado do Rio de Janeiro, Rio de Janeiro, RJ, Brazil, 4 Department of Infection Prevention and Control, AC Camargo Cancer Center, São Paulo, SP, Brazil

* mp.curado@accamargo.org.br

## Abstract

### Background

To investigate the excess of deaths by specific causes, in the first half of 2020 in the city of São Paulo-Brazil, during the COVID-19 pandemic.

### Methods

Ecological study conducted from 01/01 to 06/30 of 2019 and 2020. Population and mortality data were obtained from DATASUS. The standardized mortality ratio (SMR) by age was calculated by comparing the standardized mortality rate in 2020 to that of 2019, for overall and specific mortality. The ratio between the standardized mortality rate due to COVID-19 in men as compared to women was calculated for 2020. Crude mortality rates were standardized using the direct method.

### Results

COVID-19 was responsible for 94.4% of the excess deaths in São Paulo. In 2020 there was an increase in overall mortality observed among both men (SMR 1.3, 95% CI 1.17–1.42) and women (SMR 1.2, 95% CI 1.06–1.36) as well as a towards reduced mortality for all cancers. Mortality due to COVID-19 was twice as high for men as for women (SMR 2.1, 95% CI 1.67–2.59). There was an excess of deaths observed in men above 45 years of age, and in women from the age group of 60 to 79 years.

### Conclusion

There was an increase in overall mortality during the first six months of 2020 in São Paulo, which seems to be related to the COVID-19 pandemic. Chronic health conditions, such as cancer and other non-communicable diseases, should not be disregarded.

**Data Availability Statement:** The data underlying the results presented in the study are available from: http://tabnet.saude.prefeitura.sp.gov.br/cgi/deftohtm3.exe?secretarias/saude/TABNET/SIM_

PROV/obitop.def http://tabnet.datasus.gov.br/cgi/deftohtm.exe?popsvs/cnv/popbr.def.

**Funding:** The author(s) received no specific funding for this work.

**Competing interests:** The authors have declared that no competing interests exist.

## Introduction

In March 2020, the World Health Organization (WHO) declared the COVID-19 pandemic [1]. Since then, the disease has spread rapidly across five continents. As of June 30, 2020, more than 10 million cases and around 500,000 deaths had been reported worldwide. Brazil was responsible for more than 1.3 million of these cases and 57,000 deaths [2], while the municipality of São Paulo reported 151,414 cases and 7,123 deaths [3].

It has been well reported in the literature that people with non-communicable diseases (such as cardiovascular, hypertension, diabetes, congestive heart failure, chronic kidney disease, or cancer), males, and those of advanced age have an increased risk of death from COVID-19 [4–6]. The main causes of death worldwide are cardiovascular diseases and cancer [7]. Brazil follows this global pattern, reporting in 2015 a cardiovascular mortality rate of 256 deaths per 100,000 and a cancer mortality rate of 133.5 deaths per 100,000 [8]. In 2018 it was estimated that deaths from cancer in Brazil totaled 243,588 [9].

An excess of deaths during the pandemic has been described in several countries [10–12]. This is likely in part due to the lethality of COVID-19, but can also be a consequence of overloaded health services, resulting in poorer care for patients with chronic diseases such as cancer. In addition, economic and social effects of the pandemic likely worsened the overall health of the population and contributed to increased mortality from all causes. Therefore, investigating the effect of COVID-19 on increased mortality may help elucidate the impact of this disease on various causes of death [13].

In São Paulo, the epicenter of the pandemic in Brazil, the city established a data system in which it is possible to check the basic cause of all deaths in the municipality [14], since June 2020, making it possible to monitor and analyze the pattern of mortality during the period of the Sars-CoV-2 pandemic. Detailed knowledge of the death profile will allow the design of strategies to reduce mortality in the populations at risk. However, there have been few studies on excess overall and specific mortality during the COVID-19 pandemic, using population data in Brazil.

Therefore, this study aims to investigate the excess of deaths by underlying causes, in the first half of 2020 as compared to the same period of 2019, during the COVID-19 pandemic, in the city of São Paulo, Brazil.

## Materials and methods

This ecological study was conducted using data on overall and specific mortality, in the period from 01/01 to 06/30 of 2019 and 2020, and population data for the year 2010 from the city of São Paulo–Brazil (date of the last demographic census). The year 2019 was used as a reference, assuming that the number of deaths that occurred in 2019 would be close to that expected in 2020. Mortality data, included data on death certified COVID-19, were obtained from Mortality Information System (SIM) [14] and population data were obtained from the DATASUS [15]. Deaths in which sex and age data were unknown (0.1%) were excluded from the analysis.

The standardized mortality ratio (SMR) by age was calculated as the ratio between the standardized mortality rate of 2020 and that of 2019, for both overall and specific mortality. The ratio between the standardized mortality rate due to COVID-19 in men as compared to women was also calculated for 2020. The same analyses were also performed for each age group (0–9, 10–29, 30–44, 45–59, 60–79, 80 and over) and sex. Crude mortality rates were standardized using the direct method, using the standard "worldwide" population, created by Segi in 1960 and modified in 1966 [16]. The objective of the present analysis was to assess whether in the year 2020 there was an excess of deaths in the municipality of São Paulo, and whether mortality due to COVID-19 was higher among men.

SMRs were calculated for overall mortality and for all cancers, by age groups (0–9, 10–29, 30–44, 45–59, 60–79, 80 and over), sex and such as specific mortality: diabetes mellitus, congestive heart failure, cerebral vascular accident, acute myocardial infarction, cardiovascular diseases, and COVID-19, according to definitions in the international classification of diseases, 10th edition (ICD-10). S1 Table presents the ICD-10 codes for all categories considered in this analyses.

Confidence intervals were calculated at the 95% level (95% CI) and the Fisher's exact test with a significance level of 5%. All analyses were performed using Excel software (version 10) and OpenEpi [17].

## Results

In the first half of 2020, in the municipality of São Paulo, 23,156 deaths were reported among men and 21,782 among women, compared with 17,862 and 18,144, respectively, in the same period of 2019. This represents an increase in overall mortality of 29.6% for men and 20.0% for women. The number of deaths from COVID-19 was 34.3% higher in men (4,830) than in women (3,597). The number of deaths from all cancers decreased from 2019 to 2020, by 14.6% among men (3,762 vs. 3,211) and 10.1% among women (4,059 vs. 3,649) (Tables 1 and 2).

Among men, the excess of deaths from all causes estimated in the first half of 2020 (5,294) in the city of São Paulo was 9.6% higher than the number of deaths attributed to COVID-19 (4,830). In women, the estimated excess of deaths from all causes (3,638) was 1.1% higher than the number of deaths attributed to COVID-19 (3,597) (Tables 1 and 2).

In 2020, there was an increase in overall age-standardized mortality rates for both men (SMR 1.3, 95% CI 1.17–1.42) and women (SMR 1.2, 95% CI 1.06–1.36). Cancer age-standardized mortality rates on showed a downward trend in both men (SMR 0.9, 95% CI 0.66–1.09) and women (SMR 0.9, 95% CI 0.67–1.20), as well as, all cardiovascular diseases men (SMR 0.9, 95% CI 0.75–1.13) and women (SMR 0.9, 95% CI 0.68–1.15). Similar downward trends were observed for most specific cancer locations, except for kidney, skin melanoma, salivary gland, cervix uteri, corpus uteri, penis, lip, oral cavity (female), and pancreas (female), for which SMRs were higher than one but with non-statistically significant confidence intervals (Table 1).

There was an increase in overall age-standardized mortality rates for males for all age groups above 45 years old. However, for females an increase in overall age-standardized mortality rates was observed only in the age group of 60 to 79 years (SMR 1.3, 95% CI 1.05–1.52). There was a decrease in age-standardized mortality rates for all types of cancer in all age groups in 2020 (Table 3). However, the results of cancer mortality should be interpreted with caution, since the confidence interval included the value 1.

The age-standardized mortality rates due to COVID-19 for men in São Paulo was twice that observed for women (SMR 2.1, 95% CI 1.67–2.59). When stratifying these data by age group, there was an excess of deaths among men compared to women for all age groups from 45 years onwards (Table 2).

## Discussion

In the municipality of São Paulo, the excess of deaths in the first half of 2020 was 24.9%, and of these, 94.4% had COVID-19 as the specific mortality. There was an increase in overall mortality for both sexes, particularly in men aged 45 and over and among women aged 60–79 years.

The present findings revealed that in the municipality of São Paulo the excess of deaths from all causes was 9.6% (men) and 1.1% (women) greater than the number of deaths attributed to COVID-19. This is a small margin compared to those reported in similar studies elsewhere; in Portugal, for example, the increase in mortality was 3 to 5 times higher than that explained by

**Table 1. Standardized mortality ratios by male vs. female and overall and specific mortality in the municipality of São Paulo, Brazil, from January to June of 2019 vs. 2020.**

| | Male | | | | | Female | | | | |
|---|---|---|---|---|---|---|---|---|---|---|
| | Deaths | | | | | Deaths | | | | |
| Overall and specific mortality | 2019 | 2020 | SMR[a] | CI95%[b] | p[c] | 2019 | 2020 | SMR[a] | CI95%[b] | p[c] |
| All causes | 17862 | 23156 | 1.3 | 1.17–1.42 | <0.001 | 18144 | 21782 | 1.2 | 1.06–1.36 | 0.003 |
| Diabetes mellitus | 514 | 553 | 1.1 | 0.52–1.85 | 0.834 | 558 | 627 | 1.1 | 0.47–2.19 | 0.638 |
| All cardiovascular diseases | 5729 | 5314 | 0.9 | 0.75–1.13 | 1.000 | 5864 | 5280 | 0.9 | 0.68–1.15 | 0.703 |
| Cerebral vascular accident | 204 | 316 | 1.5 | 0.62–3.28 | 0.300 | 237 | 345 | 1.5 | 0.48–3.88 | 0.420 |
| Acute myocardial infarction | 1787 | 1599 | 0.9 | 0.60–1.26 | 0.538 | 377 | 409 | 0.8 | 0.45–1.36 | 0.476 |
| Congestive heart failure | 235 | 323 | 1,4 | 0.55–2.91 | 0.435 | 386 | 404 | 1.1 | 0.31–2.54 | 0.897 |
| All cancers | 3762 | 3211 | 0.9 | 0.66–1.09 | 0.296 | 4059 | 3649 | 0.9 | 0.67–1.20 | 1.000 |
| Lung | 497 | 480 | 1.0 | 0.44–1.70 | 0.890 | 472 | 408 | 0.9 | 0.36–1.95 | 0.727 |
| Breast | - | - | - | - | - | 85 | 82 | 0.9 | 0.47–1.81 | 0.865 |
| Colorectum | 460 | 403 | 0.9 | 0.40–1.70 | 0.720 | 478 | 440 | 0.9 | 0.38–2.03 | 0.802 |
| Prostate | 426 | 347 | 0.8 | 0.56–2.35 | 0.604 | - | - | - | - | - |
| Stomach | 336 | 254 | 0.8 | 0.26–1.66 | 0.566 | 185 | 162 | 0.9 | 0.11–2.62 | 0.874 |
| Liver | 188 | 165 | 0.9 | 0.31–2.51 | 0.847 | 163 | 124 | 0.7 | 0.11–2.75 | 0.704 |
| Oesophagus | 168 | 139 | 0.8 | 0.19–2.27 | 0.761 | 38 | 36 | 0.9 | 0.05–7.84 | 0.930 |
| Cervix uteri | - | - | - | - | - | 132 | 149 | 1.1 | 0.35–4.12 | 0.844 |
| Thyroid | 13 | 11 | 0.9 | 0.11–16.04 | 0.983 | 18 | 16 | 0.9 | 0.11–16.77 | 0.949 |
| Bladder | 129 | 115 | 0.8 | 0.10–2.35 | 0.795 | 68 | 41 | 0.6 | 0.03–4.72 | 0.734 |
| Non-Hodgkin lymphoma | 115 | 170 | 0.6 | 0.11–2.70 | 0.577 | 96 | 64 | 0.7 | 0.02–3.23 | 0.736 |
| Pancreas | 229 | 218 | 0.9 | 0.37–2.39 | 0.911 | 302 | 318 | 1.1 | 0.31–2.49 | 0.881 |
| Leukaemia | 140 | 92 | 0.6 | 0.09–2.19 | 0.551 | 118 | 92 | 0.8 | 0.16–3.89 | 0.789 |
| Kidney | 67 | 77 | 1.2 | 0.20–4.68 | 0.797 | 57 | 63 | 1.1 | 0.03–4.98 | 0.944 |
| Corpus uteri | - | - | - | - | - | 99 | 112 | 1.1 | 0.18–4.35 | 0.880 |
| Lip, oral cavity | 101 | 88 | 0.9 | 0.12–2.88 | 0.84 | 29 | 33 | 1.3 | 0.08–11.9 | 0.885 |
| Brain, central nervous system | 165 | 108 | 0.7 | 0.08–1.89 | 0.563 | 164 | 135 | 0.8 | 0.10–2.14 | 0.737 |
| Ovary | - | - | - | - | - | 187 | 162 | 0.9 | 0.25–2.98 | 0.847 |
| Melanoma of skin | 26 | 35 | 1.3 | 0.05–7.84 | 0.838 | 38 | 43 | 1.1 | 0.05–8.01 | 0.953 |
| Gallbladder | 58 | 46 | 0.8 | 0.02–3.54 | 0.836 | 95 | 80 | 0.9 | 0.02–4.84 | 0.888 |
| Larynx | 124 | 124 | 1.0 | 0.26–3.08 | 0.994 | 25 | 16 | 0.6 | 0.08–11.9 | 0.829 |
| Multiple myeloma | 68 | 52 | 0.7 | 0.02–2.92 | 0.775 | 83 | 52 | 0.7 | 0.02–3.88 | 0.742 |
| Nasopharynx | 10 | 6 | 0.7 | 0.14–20.49 | 0.906 | - | 4 | - | - | - |
| Oropharynx | 53 | 41 | 0.8 | 0.02–3.68 | 1.00 | 14 | 7 | 0.6 | 0.16–24.59 | 0.876 |
| Hypopharynx | 18 | 15 | 0.9 | 0.07–10.54 | 0.932 | 4 | 4 | 1.1 | 0.04–6.14 | 1.00 |
| Hodgkin lymphoma | 8 | 3 | 0.3 | 0.18–26.35 | 0.809 | 5 | 4 | 0.7 | 0.36–52.7 | 0.939 |
| Testis | 19 | 11 | 0.6 | 0.09–13.17 | 0.835 | - | - | - | - | - |
| Salivary glands | 9 | 10 | 1.1 | 0.16–24.59 | 0.958 | 4 | 8 | 2.1 | 0.63–92.22 | 0.764 |
| Vulva | - | - | - | - | - | 19 | 17 | 1.0 | 0.13–19.41 | 0.981 |
| Penis | 4 | 7 | 1.7 | 0.31–46.11 | 0.859 | - | - | - | - | - |

[a]SMR: Standardized Mortality Ratio

[b]CI95%: 95% Confidence Interval

[c]p: Fisher's exact test p-value

deaths from COVID-19 [10]. In Italy and Germany the excess of deaths was also greater than that officially registered by the disease [11, 12]. Another study carried out in Italy revealed an increase in mortality 60% greater than the number of deaths attributed to COVID-19 [18].

**Table 2. Standardized mortality ratios for COVID-19 by age and male vs. female in São Paulo, Brazil, from January to June 2020.**

| | Deaths | | 2020 | | |
|---|---|---|---|---|---|
| COVID-19 | Male | Female | SMR[a] | CI95%[b] | p[c] |
| All age groups | 4830 | 3597 | 2.1 | 1.67–2.59 | <0.001 |
| 0–9 | 4 | 5 | 0.7 | 0.16–23.06 | 0.900 |
| 10–29 | 47 | 30 | 1.6 | 0.03–4.72 | 0.734 |
| 30–44 | 333 | 166 | 2.2 | 0.76–4.80 | 0.072 |
| 45–59 | 881 | 501 | 2.1 | 1.24–3.47 | 0.002 |
| 60–79 | 2357 | 1537 | 2.2 | 1.66–2.93 | <0.001 |
| 80 and over | 1208 | 1358 | 1.8 | 1.09–2.80 | 0.008 |

[a]SMR: Standardized Mortality Ratio

[b]CI95%: 95% Confidence Interval

[c]p: Fisher's exact test p-value

For malignant neoplasms there was a trend towards decreased mortality. The increase in all-cause mortality during the pandemic highlights the consequences of the overload of COVID-19 on health systems [11].

A Brazilian study identified an excess of deaths from all causes among both men and women, reaching 24% in men and 14% in women [19]. In this study, the largest increases were seen in states in the North and Northeast regions. In the city of São Paulo, the month of May showed a 42% excess of deaths among men and 33% in women, results similar to those of the present study (30% and 20%, respectively).

It is noteworthy that, in this study, no cause other than COVID-19 showed an increase in deaths in 2020, but it should be expected that patients with COVID-19 will evolve with serious

**Table 3. Standardized mortality ratios by age and male vs. female in São Paulo, Brazil, from January to June of 2019 vs. 2020.**

| | Male | | | | | Female | | | | |
|---|---|---|---|---|---|---|---|---|---|---|
| | Deaths | | | | | Deaths | | | | |
| | 2019 | 2020 | SMR[a] | CI95%[b] | p[c] | 2019 | 2020 | SMR[a] | CI95%[b] | p[c] |
| **Overall Mortality** | | | | | | | | | | |
| 0–9 | 606 | 479 | 0.8 | 0.46–1.31 | 0.372 | 464 | 401 | 0.9 | 0.48–1.46 | 0.604 |
| 10–29 | 527 | 688 | 1.3 | 0.69–2.19 | 0.374 | 319 | 366 | 1.1 | 0.52–2.45 | 0.762 |
| 30–44 | 1108 | 1558 | 1.4 | 0.92–2.11 | 0.108 | 757 | 968 | 1.3 | 0.71–2.15 | 0.377 |
| 45–59 | 3176 | 4198 | 1.3 | 1.04–1.66 | 0.020 | 2104 | 2593 | 1.2 | 0.88–1.71 | 0.214 |
| 60–79 | 7677 | 10216 | 1.3 | 1.16–1.52 | <0.001 | 6510 | 8250 | 1.3 | 1.05–1.52 | 0.008 |
| 80 and over | 4768 | 6017 | 1.3 | 1.02–1.54 | 0.027 | 7990 | 9204 | 1.2 | 0.89–1.45 | 0.250 |
| **Cancer Mortality** | | | | | | | | | | |
| 0–9 | 13 | 6 | 0.5 | 0.07–10.54 | 0.761 | 12 | 11 | 0.9 | 0.07–10.85 | 0.972 |
| 10–29 | 58 | 53 | 0.9 | 0.02–3.72 | 0.935 | 62 | 40 | 0.7 | 0.02–3.58 | 0.73 |
| 30–44 | 135 | 115 | 0.8 | 0.12–2.82 | 0.814 | 245 | 240 | 1,0 | 0.34–2.75 | 0.982 |
| 45–59 | 724 | 587 | 0.8 | 0.40–1.43 | 0.51 | 817 | 689 | 0.8 | 0.43–1.54 | 0.608 |
| 60–79 | 2081 | 1807 | 0.9 | 0.63–1.19 | 0.405 | 1925 | 1751 | 0.9 | 0.61–1.34 | 0.649 |
| 80 and over | 809 | 696 | 0.9 | 0.45–1.53 | 0.628 | 998 | 918 | 0.9 | 0.39–1.83 | 0.831 |

[a]SMR: Standardized Mortality Ratio

[b]CI95%: 95% Confidence Interval

[c]p: Fisher's exact test p-value

and lethal cardiovascular outcomes [20]. In the study by Azevedo e Silva et al. [19], however, it was found that 33.5% more deaths in the country, from March to May 2020, were not registered as having been due to COVID-19. This may reflect the worsening of chronic health problems as medical monitoring was discontinued due to the pandemic.

On the other hand, we cannot rule out the hypothesis that the number of deaths with an specific mortality of COVID-19 in São Paulo may be over-registered in the mortality system. There may have been a greater trend in the notification of deaths by COVID-19 and, as a result, the other causes may have been underreported. Such excessive attribution of cause of death are expected at critical times such as the COVID-19 pandemic [21].

It has been previously described that the risk of death from COVID-19 among men is twice that in women, despite the number of cases being similar; the results of the present study are in alignment. In contrast, a study conducted in the USA did not identify any difference in mortality between the sexes [22]. It should be noted that sex differences in mortality due to COVID-19 are not consistent across all age groups. The excess of deaths due to COVID-19 reported here was identified in men aged 45 and over, which is in line with what has been reported elsewhere [23].

The excess of deaths due to COVID-19 in men is possibly associated with a greater number of comorbidities and exposure to risk factors (alcoholism, smoking, and occupational exposures) which are risk factors related to male behavior [24]. In addition, there seems to be an interaction between gender and age, with males of advanced age having the highest risk of death by COVID-19. This association should be studied further in order to understand the clinical evolution of the disease.

During the COVID-19 pandemic in the city of São Paulo, a 30% increase in overall and specific mortality was identified among men over 45 and women in the 60 to 79 age group, which is consistent with other studies [10–12]. This increase in overall mortality in elderly individuals is possibly a result of the excess of deaths related to COVID-19.

SMRs for the cancer set and for most of the cancer groups analyzed were lower not significant. A study carried out at twenty health institutions in the USA revealed that cancer incidence rates declined as COVID-19 advanced in the country. Breast, prostate, and melanoma cancer showed the greatest decline in incidence, while lung, colorectal, and hematological cancers were the least impacted [25]. It is critical to monitor the effects of the COVID-19 pandemic on the profile of cancer mortality in Brazil. Some reports have shown that the impact may be important, such as in the reduced number of oral biopsies in the Brazilian Unified Health System, especially in the southeast region (-75.6%). This may ultimately result in the diagnosis of tumors in more advanced stages and with poorer prognosis [26].

The COVID-19 pandemic has driven patients with other diseases away from health services, which impacted on the screening, diagnosis, and treatment of diseases such as cancer [27]. The results of the present study should serve as an alert for the need to implement cancer surveillance measures. It is essential to investigate how cancer patients are being assisted. Social distancing measures may on the one hand protect against exposure to the virus and other infections, but on the other hand can delay cancer treatment and surveillance.

The findings of this study illustrate the situation in the municipality of São Paulo and cannot be extrapolated to the rest of Brazil. This, the largest city in Brazil, was where SARS-CoV-2 first entered and remains the place that accumulates the largest number of COVID-19 cases in the country. The strength of this study is in its use of official mortality data provided by the DATASUS Mortality Information System (SIM) and the finding that they offer powerful tools for the implementation of health surveillance in monitoring excess of deaths in Brasil.

In conclusion, in order to avoid an excess of deaths in the coming months, it is necessary for health systems to remain alert and prepared for future peaks of COVID-19. The increase in

overall mortality during the pandemic period seems to be related to Sars-CoV-2. However, likely that a share of the deaths classified as COVID-19 may also have a diagnosis of cancer and cardiovascular diseases listed in the death certificate, and this may account for the apparent decrease in cancer and cardiovascular diseases mortality. It is suggested that individuals with comorbidities associated with risk of death by COVID-19 pursue preventive actions and continuous treatment adjustments, as many uncertainties about COVID-19 still remain.

## Supporting information

**S1 Table. Causes of deaths and corresponding ICD-10 codes.**
(DOCX)

## Author Contributions

**Conceptualization:** Gisele Aparecida Fernandes, Antonio Paulo Nassar Junior, Maria Paula Curado.

**Data curation:** Gisele Aparecida Fernandes.

**Formal analysis:** Gisele Aparecida Fernandes, Maria Paula Curado.

**Investigation:** Gisele Aparecida Fernandes, Maria Paula Curado.

**Methodology:** Gisele Aparecida Fernandes, Maria Paula Curado.

**Project administration:** Gisele Aparecida Fernandes, Maria Paula Curado.

**Supervision:** Maria Paula Curado.

**Writing – original draft:** Gisele Aparecida Fernandes, Maria Paula Curado.

**Writing – review & editing:** Gisele Aparecida Fernandes, Antonio Paulo Nassar Junior, Gulnar Azevedo e Silva, Diego Feriani, Ivan Leonardo Avelino França e Silva, Pedro Caruso, Maria Paula Curado.

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
