## [Decision Letter · Decision Letter 0]

24 Feb 2021

PONE-D-21-00780

Excess mortality by underlying causes of deaths in the city of São Paulo, Brazil, during the COVID-19 pandemic

PLOS ONE

Dear Dr. Curado,

Thank you for submitting your manuscript to PLOS ONE. After careful consideration, we feel that it has merit but does not fully meet PLOS ONE’s publication criteria as it currently stands. Therefore, we invite you to submit a revised version of the manuscript that addresses the points raised during the review process.

The manuscript has been sent out to an outstanding reviewer with a strong background and competence in the study of cause-specific excess mortality. The manuscript is interesting but there are several issues that should be addressed. One of the major comments raised up by the reviewer is related to the authors’ choice to include hypertension as a cause of deaths, without considering chronic cardiovascular conditions, leaving aside an important share of cardiovascular deaths. Please find attached the comments.

In addition to the reviewers’ comments, I would ask to:

Clarify if the DATASUS website (page 4 line 100) also included data on certified COVID-19 mortality.Clarify Table 1 and Table 2 by putting in a top row the label “Male” or “Female”;Specify within Table 3 that the SMR is computed for Male vs Female, and not “by gender” as specified in the table caption.

We look forward to receiving your revised manuscript.

Kind regards,

Matteo Rota, Ph.D.

Academic Editor

PLOS ONE

2. In your methods section, please provide methods for the specific statistical tests used to assess significance of your results.

3. Thank you for providing the date(s) when patient medical information was initially recorded. Please also include the date(s) on which your research team accessed the databases/records to obtain the retrospective data used in your study.

Reviewers' comments:

Reviewer's Responses to Questions

**Comments to the Author**

1. Is the manuscript technically sound, and do the data support the conclusions?

Reviewer #1: Yes

2. Has the statistical analysis been performed appropriately and rigorously? 

Reviewer #1: Yes

3. Have the authors made all data underlying the findings in their manuscript fully available?

Reviewer #1: Yes

4. Is the manuscript presented in an intelligible fashion and written in standard English?

Reviewer #1: Yes

5. Review Comments to the Author

Reviewer #1: In this manuscript Dr Fernandes et al. presented the results of an ecologic study showing the differences in cause-specific mortality between the first half of the 2020 and 2019 in the city of Sao Paulo, Brazil. They found an excess in overall mortality in 2020 that they attributed to the Covid-19 pandemic.

The topic is interesting and the manuscript is well written. However, I have some points that I think need to be addressed.

1. Line 97: It is not clear why the authors needed the 2010 population data to compute mortality rates over the first half of 2019 and 2020.

2. Line 104. I would say “…overall and specific mortality”. Mortality statistics are usually based on the underlying cause of death.

3. Line 113-117. The selection of the causes of death is quite unusual and I disagree with the choices made by the authors. In particular, I would not include hypertension since it is a risk factor rather than an underlying cause of death, and it is selected as an underlying cause of death only when there are no other more specific causes mentioned on the death certificate, such as cerebrovascular and renal diseases. Moreover, I cannot understand why the authors did not consider chronic cardiovascular conditions, leaving aside an important share of cardiovascular deaths. Finally, I do not think it is worth reporting data on obesity, since it is rarely mentioned on death certificates.

4. Results. Text and tables are difficult to follow since tables are not commented in the order they were presented.

5. Results. When the authors refer to an increase or decrease in mortality they should specify if they refer to a change in the absolute numbers or age-standardized rates.

6. Table 1. Some SMR are emphasized in bold, with apparent no reason, or at least I did not get it. In any case, I would avoid it and leave the reader judge the relevance of the result.

7. Table 2. How did the authors compute the age-standardized mortality ratio by age groups if the age groups presented have the same wide of those used to get the weights from the standard population? Please clarify.

8. Table 2. Please avoid bold emphasis.

9. Table 3. See the above comment on age groups.

10. Lines 205-206. The increase in overall mortality was also observed in younger men, aged 30-44 years, and among women aged 60-79 years. Please revise the sentence accordingly.

11. Conclusions: The authors concluded that the increase in overall mortality during the pandemic period is related to SARS-CoV-2 and other health conditions such as cancer. However, cancer deaths in Sao Paolo decreased from 3762 in 2019 to 3211 in 2020. Thus, the pandemic does not seem to have directly increased cancer mortality. It is, however, likely that a share of the deaths classified as COVID-19 may also have a diagnosis of cancer listed in the death certificate, and this may account for the apparent decrease in cancer mortality.

6. PLOS authors have the option to publish the peer review history of their article (what does this mean?). If published, this will include your full peer review and any attached files.

Reviewer #1: No

---

## [Author Response · Author response to Decision Letter 0]

22 Apr 2021

Reviewers' comments:

Manuscript Number: PONE-D-21-00780

Title: Excess mortality by specific causes of deaths in the city of São Paulo, Brazil, during the COVID-19 pandemic

Response to Editor

Point 1: Clarify if the DATASUS website (page 4 line 100) also included data on certified COVID-19 mortality.

Response 1: Thank you for your comment. Data on certified COVID-19 mortality were obtained from the SIM. We included the following sentence in line 102: “Mortality data, included data on death certified COVID-19, were obtained from SIM [14] and population data were obtained from the DATASU [15].”

Point 2: Clarify Table 1 and Table 2 by putting in a top row the label “Male” or “Female”

Response 2: Thank you for your suggestion. The changes were made to the tables.

Point 3: Specify within Table 3 that the SMR is computed for Male vs Female, and not “by gender” as specified in the table caption

Response 3: Thank you for your suggestion. The changes were made to the tables caption.

Response to Reviewer 1 Comments

Point 1: Line 97: It is not clear why the authors needed the 2010 population data to compute mortality rates over the first half of 2019 and 2020.

Response 1: Thank you for your comment. We used population data from 2010 to calculate mortality rates in the first half of 2019 and 2020, because it was the date of the last demographic census conducted in Brazil. We included the following sentence in line 97 and 98: “(date of the last demographic census)”

Point 2: Line 104. I would say “…overall and specific mortality”. Mortality statistics are usually based on the underlying cause of death.

Response 2: Thank you for your comment. We deleted the sentence “due to underlying cause” from the manuscript.

Point 3: Line 113-117. The selection of the causes of death is quite unusual and I disagree with the choices made by the authors. In particular, I would not include hypertension since it is a risk factor rather than an underlying cause of death, and it is selected as an underlying cause of death only when there are no other more specific causes mentioned on the death certificate, such as cerebrovascular and renal diseases. Moreover, I cannot understand why the authors did not consider chronic cardiovascular conditions, leaving aside an important share of cardiovascular deaths. Finally, I do not think it is worth reporting data on obesity, since it is rarely mentioned on death certificates.

Response 3: Thank you for your comment. We excluded mortality from obesity and arterial hypertension from the article and included mortality from congestive heart failure and all cardiovascular diseases.

Point 4: Results. Text and tables are difficult to follow since tables are not commented in the order they were presented

Response 4: Thank you for your comment. We organized the tables in the order in which they were commented on in the text.

Point 5: Results. When the authors refer to an increase or decrease in mortality they should specify if they refer to a change in the absolute numbers or age-standardized rates.

Response 5: Thank you for your comment. We refer to the age-standardized mortality rates. We included this information all over the manuscript to clarify.

Point 6: Table 1. Some SMR are emphasized in bold, with apparent no reason, or at least I did not get it. In any case, I would avoid it and leave the reader judge the relevance of the result.

Response 6: Thank you for your comment. We excluded the emphasized in bold the Table 1.

Point 7: Table 2. How did the authors compute the age-standardized mortality ratio by age groups if the age groups presented have the same wide of those used to get the weights from the standard population? Please clarify.

Response 7: Thank you for your comment. First we calculate the crude mortality rate for all age groups in 2019 and 2020, stratified as described in the method session. The crude mortality rate result was multiplied by age groups from the standard Segi population. The SMR was calculated by dividing the standardized mortality rate by 2020 age group by the standardized mortality rate by the same age group of 2019, in order to compare the differences that occurred. We included the following sentence in lines 115, 116 and 117: “SMRs were calculated for overall mortality and for all cancers, by age groups (0–9, 10–29, 30–44, 45–59, 60–79, 80 and over), sex and such as specific mortality:”

Point 8 : Table 2. Please avoid bold emphasis.

Response 8 : Thank you for your comment. We excluded the emphasized in bold.

Point 9 : Table 3. See the above comment on age groups.

Response 9 : Please, see answer in response 7 above.

Point 10 : Lines 205-206. The increase in overall mortality was also observed in younger men, aged 30-44 years, and among women aged 60-79 years. Please revise the sentence accordingly

Response 10 : Thank you for your comment. We included the following sentence in line 212: “and among women aged 60-79 years”

Point 11 : Conclusions: The authors concluded that the increase in overall mortality during the pandemic period is related to SARS-CoV-2 and other health conditions such as cancer. However, cancer deaths in Sao Paolo decreased from 3762 in 2019 to 3211 in 2020. Thus, the pandemic does not seem to have directly increased cancer mortality. It is, however, likely that a share of the deaths classified as COVID-19 may also have a diagnosis of cancer listed in the death certificate, and this may account for the apparent decrease in cancer mortality. 

Response 11: Thank you for your comment. We added part of your comment to the article in lines 294, 295, 296,297 and 298: However, likely that a share of the deaths classified as COVID-19 may also have a diagnosis of cancer and cardiovascular diseases listed in the death certificate, and this may account for the apparent decrease in cancer and cardiovascular diseases mortality.

---

## [Decision Letter · Decision Letter 1]

12 May 2021

Excess mortality by underlying causes of deaths in the city of São Paulo, Brazil, during the COVID-19 pandemic

PONE-D-21-00780R1

Dear Dr. Curado,

We’re pleased to inform you that your manuscript has been judged scientifically suitable for publication and will be formally accepted for publication once it meets all outstanding technical requirements.

Kind regards,

Matteo Rota, Ph.D.

Academic Editor

PLOS ONE

Additional Editor Comments (optional):

Reviewers' comments:

Reviewer's Responses to Questions

**Comments to the Author**

1. If the authors have adequately addressed your comments raised in a previous round of review and you feel that this manuscript is now acceptable for publication, you may indicate that here to bypass the “Comments to the Author” section, enter your conflict of interest statement in the “Confidential to Editor” section, and submit your "Accept" recommendation.

Reviewer #1: All comments have been addressed

2. Is the manuscript technically sound, and do the data support the conclusions?

Reviewer #1: Yes

3. Has the statistical analysis been performed appropriately and rigorously? 

Reviewer #1: Yes

4. Have the authors made all data underlying the findings in their manuscript fully available?

Reviewer #1: Yes

5. Is the manuscript presented in an intelligible fashion and written in standard English?

Reviewer #1: Yes

6. Review Comments to the Author

Reviewer #1: Thank you for addressing my comments.

I have only one further minor point: the authors emphasized in the abstract that mortality from cancer tended to be lower in 2020 than in 2019. However, similar results were found for CVD deaths, although none of the differences were statistically significant. Therefore, I rather would say that there were not significant differences in cause specific mortality.

7. PLOS authors have the option to publish the peer review history of their article (what does this mean?). If published, this will include your full peer review and any attached files.

Reviewer #1: No

---

## [Editor Report · Acceptance letter]

28 May 2021

PONE-D-21-00780R1 

Excess mortality by specific causes of deaths in the city of São Paulo, Brazil, during the COVID-19 pandemic 

Dear Dr. Curado:

I'm pleased to inform you that your manuscript has been deemed suitable for publication in PLOS ONE. Congratulations! Your manuscript is now with our production department. 

Kind regards, 

on behalf of

Dr. Matteo Rota 

Academic Editor

PLOS ONE